# Selective far-field addressing of coupled quantum dots in a plasmonic nanocavity

Jianwei Tang [1], Juan Xia [1], Maodong Fang[2], Fanglin Bao[2], Guanjun Cao[2], Jianqi Shen[1], Julian Evans[1] & Sailing He [1,2,3]

Plasmon–emitter hybrid nanocavity systems exhibit strong plasmon–exciton interactions at the single-emitter level, showing great potential as testbeds and building blocks for quantum optics and informatics. However, reported experiments involve only one addressable emitting site, which limits their relevance for many fundamental questions and devices involving interactions among emitters. Here we open up this critical degree of freedom by demonstrating selective far-field excitation and detection of two coupled quantum dot emitters in a U-shaped gold nanostructure. The gold nanostructure functions as a nanocavity to enhance emitter interactions and a nanoantenna to make the emitters selectively excitable and detectable. When we selectively excite or detect either emitter, we observe photon emission predominantly from the target emitter with up to 132-fold Purcell-enhanced emission rate, indicating individual addressability and strong plasmon–exciton interactions. Our work represents a step towards a broad class of plasmonic devices that will enable faster, more compact optics, communication and computation.

[1] Centre for Optical and Electromagnetic Research, State Key Laboratory of Modern Optical Instrumentation, National Engineering Research Center of Optical Instrumentation, JORCEP, College of Optical Science and Engineering, Zhejiang University, 310058 Hangzhou, China. [2] Centre for Optical and Electromagnetic Research, ZJU-SCNU Joint Center of Photonics, South China Academy of Advanced Optoelectronics, South China Normal University, 510006 Guangzhou, China. [3] Department of Electromagnetic Engineering, School of Electrical Engineering, Royal Institute of Technology, S-100 44 Stockholm, Sweden. These authors contributed equally: Jianwei Tang, Juan Xia, Maodong Fang. Correspondence and requests for materials should be addressed to S.H. (email: sailing@kth.se)

Plasmonic nanostructures are nanocavities with ultra-small mode volumes capable of mediating extremely strong self[1–13] and mutual[14–24] emitter interactions with large bandwidth and rich topologies at the deep subwavelength scale. Plasmonic nanostructures are also well known as efficient nanoantennas, capable of tailoring the excitation[1,2,6,25] and radiation[5,26–30] of single emitters, providing rich degrees of freedom for system addressing. Owing to these superior properties, plasmon–emitter hybrid nanosystems hold great promise as testbeds and building blocks for quantum optics and informatics[31,32]. Notably, they are currently the only room-temperature system to reach the strong coupling regime at the single-emitter level[9–11,13].

Despite remarkable strides made in the construction of emitter–plasmon hybrid nanosystems, all of the reports we are aware of involve only one addressable emitting site, which limits their relevance for a wide range of fundamental experiments and devices involving interactions among emitters[18–24,33–35]. Moving one step forward to involve more than one addressable emitter in a plasmon–emitter hybrid nanosystem is urgent but a significant challenge of both nanofabrication and design.

Here we open up this critical degree of freedom by demonstrating that in a properly designed plasmon–emitter hybrid nanosystem the coupled emitters can be selectively excited and detected from the far field. To this end, two silica-encapsulated colloidal quantum dots (QDs) are employed as the emitters and precisely coupled to a U-shaped gold nanostructure that is designed to function as a combination of nanocavity and nanoantenna. As a nanocavity, the plasmonic nanostructure can enhance the emitter interactions, while as a nanoantenna it can make the emitters selectively excitable and detectable. Both selective excitation and selective detection are experimentally demonstrated with a high selectivity around 0.96 (defined as $(I_\mathrm{s} - I_\mathrm{n})/(I_\mathrm{s} + I_\mathrm{n})$, where $I_\mathrm{s}$ is the signal from the emitter to be selectively excited or detected and $I_\mathrm{n}$ is the signal from the other emitter). The emission rates of both QD emitters in the nanosystem are strongly Purcell-enhanced (by ~45-fold for Q1 and ~132-fold for Q2), which indicates that both emitters strongly couple to the plasmonic modes.

## Results

**Construction of the hybrid nanosystem.** The designed nanosystem consists of two silica-encapsulated colloidal QD emitters (Q1 and Q2) and three colloidal gold nanorods (GNRs; G1, G2 and G3) assembled into a U shape on a silica glass substrate (Fig. 1a) using atomic force microscopy (AFM) nanomanipulation (Methods). Figure 1b shows an AFM topographic image of the fabricated nanosystem. The constituent QDs Q1 and Q2 have similar emission spectra with a central wavelength of ~808 nm and similar excitation spectra (Fig. 1c). Clear blinking behaviours (random switching between on and off states) can be identified in their emission intensity time trajectories (Supplementary Fig. 1a, b), which is characteristic of single QDs[36]. The GNRs G1, G2 and G3 have similar plasmonic responses with a resonance wavelength of ~715 nm (Fig. 1c). See Supplementary Note 1 for detailed description of the structure parameters.

**Operating mechanisms.** Owing to the topology of the nanostructure, both $x$- and $y$-polarized illuminations predominantly generate $y$-oriented enhanced local fields at the QDs (Supplementary Fig. 5). This allows us to selectively excite each QD (by enhancing the excitation of one QD while suppressing the excitation of the other QD) through local field interference using elliptically polarized excitation light that defines the amplitude ratio ($\tan \theta$) and phase difference ($\varphi$) between the $y$- and $x$-

components of the excitation (Supplementary Note 2). Theoretically, for illumination with 740 nm wavelength, if its polarization is the anti-clockwise ellipse shown in the bottom-left corner of Fig.1d, the local field is strongly suppressed (~0.086-fold of the local field without the gold nanostructure) at Q2, whereas strongly enhanced (~150-fold) at Q1; if its polarization is the clockwise ellipse shown in the bottom-left corner of Fig.1e, the local field is strongly suppressed (~0.046-fold) at Q1, whereas strongly enhanced (~200-fold) at Q2. This combination of enhancement and suppression yields an excitation ratio of ~1700 (~4300), corresponding to a selectivity of ~0.9989 (~0.9995) for selective excitation of Q1 (Q2).

By placing an excited emitter at a location with tailored local density of photon states (LDOS), the decay channels of the emitter's excitonic energy can be engineered[37]. Here the plasmonic modes of the gold nanostructure provide high LDOS at the location of the QDs. Because the LDOS at Q1 (Q2) is almost exclusively associated with the plasmonic mode shown in Fig. 1f (g), the excitonic energy of Q1 (Q2) transfers almost exclusively to that plasmonic mode. The mode profile shows that the plasmonic mode is confined to a small volume that contains both QDs, which enhances both the self and mutual interactions for the QDs. Without coherence between the QDs in the current experimental system, the enhanced self interaction enhances spontaneous emission rate (known as the Purcell effect[38]; Supplementary Note 3)[1–8], while the enhanced mutual interaction enhances Förster energy transfer rate between the QDs (Supplementary Note 4)[14–17]. Although the energy transfer rate is theoretically expected to be enhanced in the nanosystem, it is still much smaller than the enhanced spontaneous emission rates, and therefore, considering the competition between the energy transfer and the spontaneous emission of the donor[16], the energy transfer efficiency is so low that we can neglect the energy transfer in the experiment (Supplementary Note 4).

Since the excitonic energy of the emitter is transferred almost exclusively to the plasmonic mode, the radiation characteristics of the emitter is determined by the radiation characteristics of the plasmonic mode. This enables radiation engineering with plasmonic nanostructures[5,26–30]. Because Q1 and Q2 transfer their excitonic energy to their respective plasmonic modes that are distinct from each other due to the QDs' distinct locations in the nanosystem (comparing between Fig. 1f and Fig. 1g), their far-field radiations should have distinct characteristics. The three constituent GNRs G1, G2 and G3 behave like three linearly polarized electric dipoles with respective orientations, amplitudes and phases (see the electric displacement vectors in Fig. 1f, g), which combine to form an elliptically polarized effective electric dipole, with its $x$-component contributed by G3 and its $y$-component contributed by G1 and G2. For both modes in Fig. 1f, g, since the GNRs are similar and the emitter is in the middle of the gap between the two nearby orthogonally oriented GNRs, the amount of energy coupled from the emitter to the two nearby GNRs should be similar, and therefore the $x$- and $y$-components of the effective electric dipole should have similar amplitudes. The key distinction between the two modes in Fig. 1f, g is the phase relation between the $x$- and $y$-components of the effective electric dipole. The $x$- and $y$-components of the effective electric dipole in Fig. 1f (g) are roughly in-phase (anti-phase) and of similar amplitude, which should therefore combine to produce a nearly linearly polarized effective electric dipole oriented at an angle of around 45° (135°). These two roughly orthogonal effective electric dipoles should produce far-field radiations with roughly orthogonal polarizations. Through far-field projections of the simulated near fields of the respective plasmonic modes, the polarization states of the far-field radiations from Q1 and Q2 can be precisely obtained as shown in Fig. 1h, i. The far-field

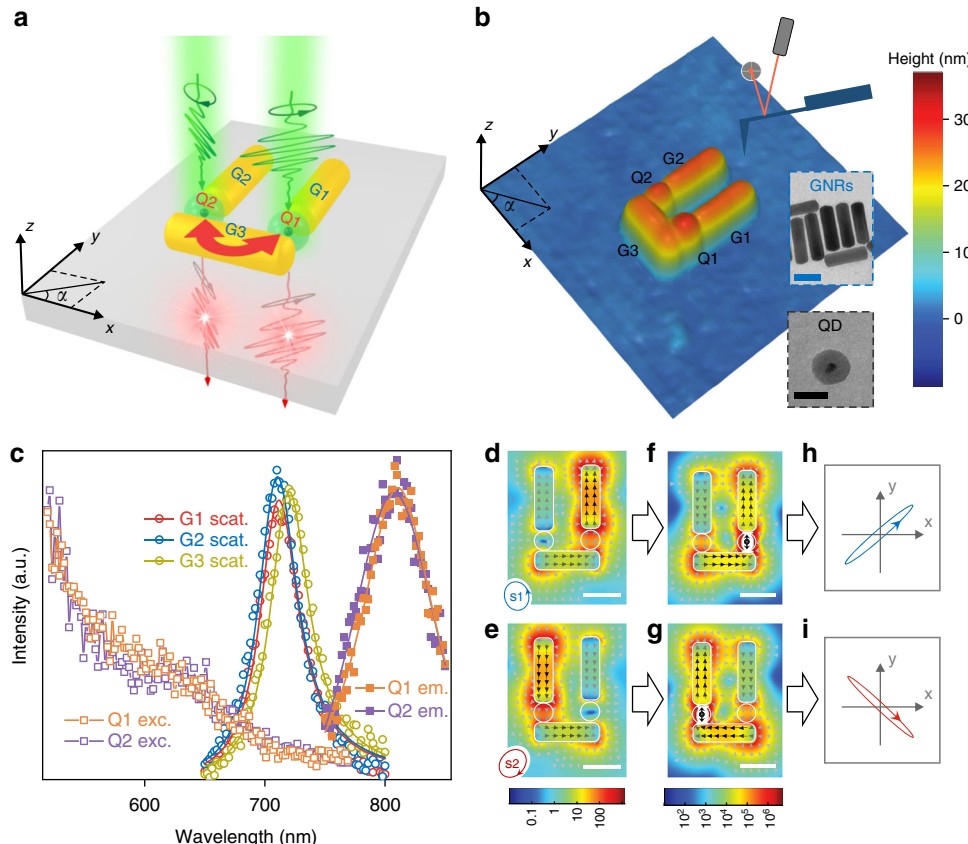

**Fig. 1** Structure and working mechanisms of the hybrid nanosystem. **a** Schematics of the structure and the main mechanism for simultaneous implementation of far-field selective excitation, plasmon-enhanced emitter interaction and far-field selective detection of the QDs. The GNRs (G1, G2 and G3) have a similar diameter of ~25 nm and a similar length of ~85 nm. The silica-encapsulated CdSeTe/ZnS core-shell QDs (Q1, Q2) have a similar diameter of ~26 nm. The gap width is ~29 nm between G1 and G3 and ~26 nm between G2 and G3. See Supplementary Note 1 for detailed structural parameters. The green pulses illustrate the excitation light with specific elliptical polarizations to selectively excite each QD. The red pulses illustrate the photons emitted from each QD with plasmon-tailored polarizations (nearly linear and roughly orthogonal). The red double arrow illustrates the plasmon-mediated interaction between the QDs. **b** AFM topographic image of the fabricated nanosystem over an area of 394 × 394 nm². Insets: Typical TEM images of GNRs (scale bar, 50 nm) and silica-encapsulated QDs (scale bar, 30 nm). **c** Measured darkfield scattering spectra (solid curves are simulated spectra) of G1, G2 and G3; measured excitation/absorption and emission (solid curves are Lorentz fits) spectra of Q1 and Q2. **d, e** Simulated maps (scale bar, 50 nm) of electric field intensities and electric displacement vectors when illuminated by a plane wave at 740 nm wavelength with elliptical polarizations s1 (**d**) and s2 (**e**) indicated in the bottom-left corner of the panel. **f, g** Simulated maps of electric field intensities and electric displacement vectors of the plasmonic mode excited at 808 nm wavelength by a y-oriented dipole at Q1 (**f**) and Q2 (**g**). **h, i** Simulated polarization states (shown by the polarization ellipses) of the emissions from Q1 (**h**) and Q2 (**i**). The emission from Q1 (Q2) is linearly polarized with a degree of linear polarization (DOLP) of 0.98 (0.986) and a polarization angle of 41° (137°)

radiation from Q1 (Q2) is linearly polarized with a degree of linear polarization (DOLP) of 0.98 (0.986) and a polarization angle of 41° (137°). Since the far-field radiations from Q1 and Q2 are nearly linearly polarized and their polarization angles are roughly orthogonal, we can selectively detect the emission from either QD with high transmittance by simply blocking the emission from the other QD using a linear polarizer. Generally, it is possible to selectively detect emitters with high transmittance if their emissions have pure polarizations well separated on the Poincaré sphere.

**Experimental demonstration**. To perform far-field selective excitation and detection, the key task is to experimentally find, for each QD, the optimal polarization for excitation suppression and optimal polarizer angle for emission blocking. We first solve these two tasks when only Q1 is in the nanosystem (before Q2 is moved into the nanosystem; Fig. 2a inset). We find the optimal excitation polarization by successively searching the elliptical polarization parameters $\varphi$ and $\theta$ to minimize the emission intensity. When we

excite with the polarization optimized at a specified wavelength (e.g. 730, 740 and 760 nm; Fig. 2a), the measured excitation enhancement spectrum shows a Fano-like dip at the desired wavelength (Fig. 2a). The non-vanishing enhancement factors at the dips (~2.9, ~2.3 and ~1.8) are attributed to imperfections in the fabrication and measurement. Since the local fields are strongly enhanced for both x- and y-polarized excitations, small imperfections can lead to significant changes in the enhancement factors. To find the optimal polarizer angle for emission blocking, we record the detected emission intensity vs. the polarizer angle. The measurement shows a high DOLP of ~0.96, in stark contrast to the partially polarized emission before Q1 couples to the gold nanostructure (Fig. 2b).

When both QDs are in the nanosystem (Fig. 1b), determining the optimal polarization for selective excitation and the optimal polarizer angle for selective detection is non-trivial due to their mutual dependence. To find the optimal excitation polarization to suppress either QD, we have to selectively detect the QD to minimize its emission intensity. To find the optimal polarizer angle to selectively detect either QD, we have to selectively excite

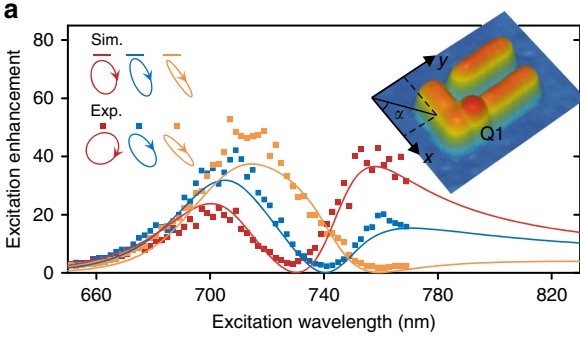

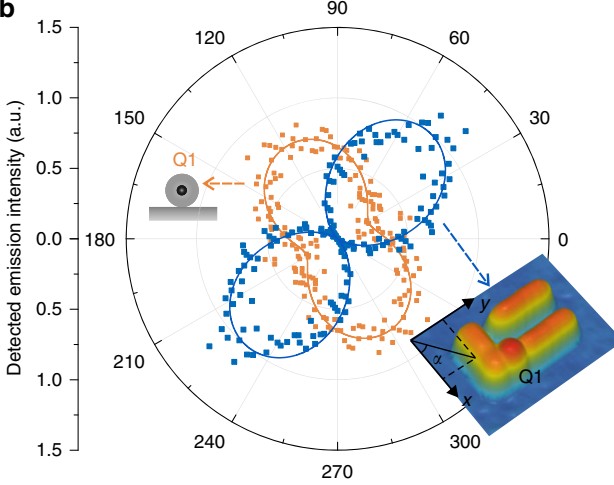

**Fig. 2** Experimental control of the excitation spectra and emission polarization with only Q1 in the nanosystem. **a** Excitation enhancement spectra when we excite with three different elliptical polarizations (indicated by the ellipses in the upper-left corner of the panel) optimized for excitation suppression at 730 nm (red), 740 nm (blue) and 760 nm (orange). **b** Polar plots of the emission polarization measurements (detected intensity vs. the angle of the linear polarizer) for Q1 before (orange data points) and after (blue data points) it couples to the gold nanostructure. The blue continuous curve is the simulated result while the orange continuous curve is a sinusoidal fit

the QD to measure its emission polarization. We overcome the mutual dependence by iteratively optimizing the excitation polarization and the polarizer angle until stable values emerge after a small number of iterations (Supplementary Note 5). When we excite with the polarization e1 (e2) optimized at 740 nm wavelength, the measured excitation enhancement spectrum for Q2 (Q1) shows a Fano-like dip at 740 nm while Q1 (Q2) remains strongly enhanced (Fig. 3a), which enables selective excitation of Q1 (Q2) at 740 nm. The measured polarizations of the emissions from the selectively excited QDs are nearly linear and roughly orthogonal (Fig. 3b), which facilitates far-field selective detection. Selective excitation can be realized in a broad spectral range (Supplementary Note 2). Already with the knowledge of the emission polarizations, we can directly find the optimal excitation polarizations for selective excitations at any other wavelength (e.g. 760 nm; Fig. 3c) without iterative optimization. The excitation enhancement factors for Q1 and Q2 change sinusoidally with $\varphi$ (Fig. 3d). We can tune the excitation ratio ranging from equal excitation to selective excitation through setting $\theta$ and $\varphi$ according to Fig. 3e. The measured excitation selectivities here (~0.92 for selective excitation of Q1 using polarization e1; ~0.93 for selective excitation of Q2 using polarization e2) are below the physical values. When Q1 is optimally suppressed with

polarization e2, Q2 is strongly enhanced by ~150-fold. Therefore, when we measure the excitation enhancement factor for the optimally suppressed Q1, the leaked part (due to the finite detection selectivity of ~0.96 estimated according to the DOLP measured when only Q1 is in the nanosystem) of the emission from the strongly enhanced Q2 can contribute a factor of ~3 to the measured enhancement factor (~5.7), meaning that the excitation enhancement factor for Q1 is actually ~2.7, which is consistent with the value measured when only Q1 is in the nanosystem (~2.3; Fig. 2a). Taking this consideration, the physical excitation selectivities are ~0.964. Similarly, taking into consideration the finite excitation selectivities (~0.964), the measured DOLPs for selectively excited Q1 and Q2 (~0.93; Fig. 3b) are also below the physical values (~0.96, consistent with the DOLP measured when only Q1 is in the nanosystem).

Lifetime analyses (Fig. 4) further confirm selective excitation and detection. When Q1 (Q2) is selectively excited with polarization e1 (e2), the measured lifetime curve (blue (red) solid data points in Fig. 4) is nearly mono-exponential, with a remarkably reduced lifetime of ~6.47 ns (~1.88 ns) compared with the intrinsic lifetime of ~291 ns (~248 ns). The fitting for the lifetime curve of selectively excited Q1 (Q2) reveals a minor ~3.4% (~4.8%) decay component with a lifetime of ~1.88 ns (~6.47 ns), which is just the lifetime of the other QD (Supplementary Note 6). Under selective excitation, the photon emission is predominantly from the target QD and the remainder is from the other QD due to the finite excitation selectivity. When Q1 and Q2 are equally excited with the excitation polarization e3 (Fig. 3d,e), the lifetime curve (yellow-green solid data points in Fig. 4) fits to a bi-exponential decay, indicating ~54% photons from Q1 (lifetime ~6.47 ns) and ~46% photons from Q2 (lifetime ~1.88 ns). Under this equal excitation, when we selectively detect only the photons from Q1 or Q2 (by setting the polarizer angle according to Fig. 3b), the nearly mono-exponential decay behaviours recover (blue and red hollowed data points in Fig. 4). The selectively detected photon emission is predominantly (~97.3% when Q1 is selectively detected; ~96.4% when Q2 is selectively detected) from the target QD and the remainder is from the other QD due to the finite detection selectivity. Since the lifetime curves are nearly mono-exponential under selective excitation or selective detection and the minor decay component can be attributed to the contribution from the other QD due to the finite selectivity, the decay of each QD can be regarded as mono-exponential. When only Q1 is in the nanosystem, the measured lifetime curve is indeed mono-exponential (Supplementary Fig. 9a). The mono-exponential decay behaviours are expected in our measurement for two reasons. First, the QDs are weakly excited, so the probability of excitation of biexcitons or multiexcitons can be neglected and the measured decay dynamics is of monoexcitons. Second, at room temperature the decay dynamics is still much slower than thermalization, so the decay dynamics can be well described with an effective decay rate[39].

Here we observe strong Purcell effects for both QDs from the reduced emission lifetimes, indicating strong light–emitter interactions. The Purcell factor (the enhancement in the total decay rate) for Q2 is ~132 ± 8, which approaches the theoretical maximum for an optimally oriented 2D transition dipole (Supplementary Note 3). Such high Purcell factors beyond 100 are rarely reported for single QDs[7,8,40] and other single emitters[2,4,6]. We achieve this high Purcell factor using deterministic plasmon-emitter coupling and with unambiguous mono-exponential exciton decay behaviour. The Purcell factor for Q1 is ~45 ± 3, which is lower than that for Q2 due to the larger gap at Q1 than at Q2 and the non-optimal orientation of Q1 (Supplementary Note 3). The strong Purcell effects, along with the plasmon-tailored emission polarizations consistent with

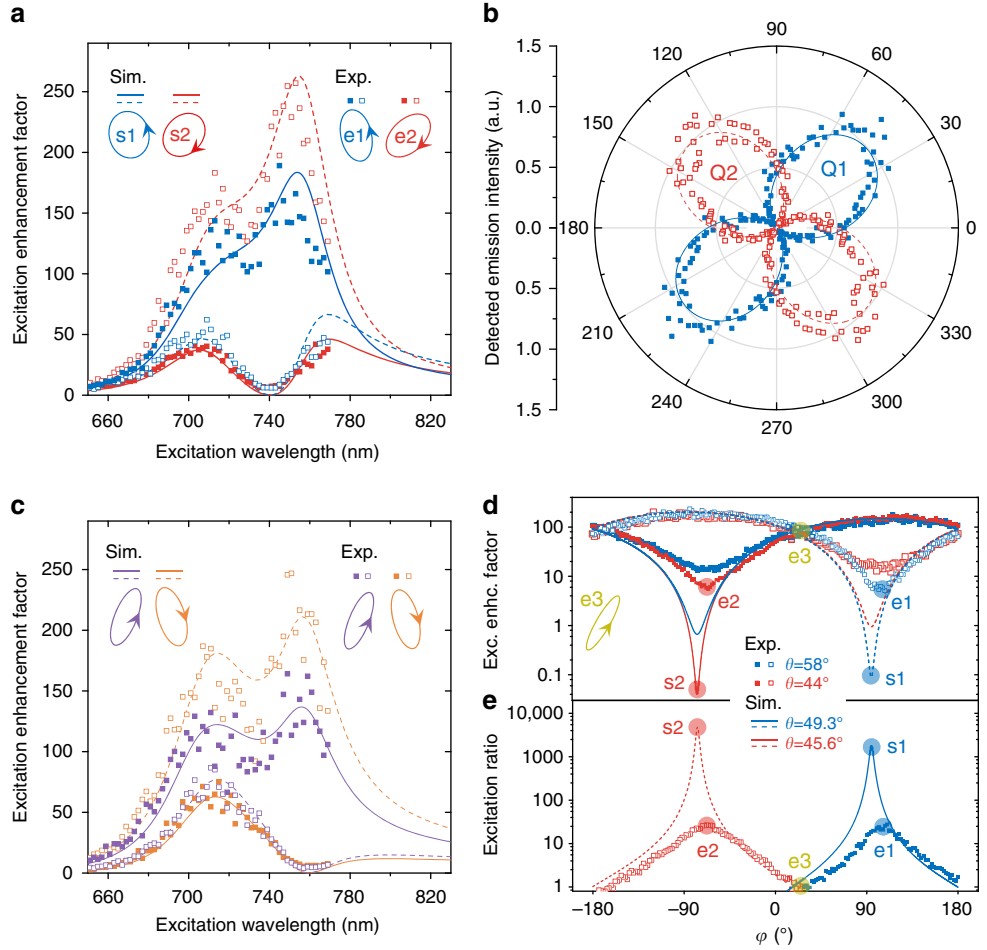

**Fig. 3** Experimental demonstration of selective far-field excitation and detection with both Q1 and Q2 in the nanosystem. **a** Excitation enhancement spectra for Q1 (solid data points; in order to measure for Q1, Q1 is selectively detected) and Q2 (hollowed data points; in order to measure for Q2, Q2 is selectively detected) when excited with elliptical polarizations for optimal selective excitation of Q1 (blue data points; elliptical polarization e1) or Q2 (red data points; elliptical polarization e2) at 740 nm wavelength. Solid (dashed) curves are simulated excitation enhancement spectra for Q1 (Q2). **b** Polar plot of the emission polarization measurement when Q1 (blue solid data points) is selectively excited (using elliptical polarization e1) or when Q2 (red hollowed data points) is selectively excited (using elliptical polarization e2). The solid and dashed curves are simulated results. **c** Same as **a** but with excitation polarizations optimized at 760 nm wavelength. **d** Excitation enhancement factor at 740 nm wavelength as a function of $\varphi$ (at two fixed $\theta$ values). Solid data points (experiment) and solid curves (simulation) are for Q1, while hollowed data points (experiment) and dashed curves (simulation) are for Q2. **e** Excitation ratio at 740 nm wavelength as a function of $\varphi$ (at two fixed $\theta$ values). Solid data points (experiment) and solid curves (simulation) are the excitation ratio $\sigma_{Q1}/\sigma_{Q2}$, while hollowed data points (experiment) and dashed curves (simulation) are the excitation ratio $\sigma_{Q2}/\sigma_{Q1}$. The blue (red) circles in **d** and **e** indicate the polarizations for optimal selective excitation of Q1 (Q2) shown in **a**. The yellow-green circles indicate the polarization e3 (inset at the left of **d**) to equally excite Q1 and Q2

simulation, indicate that the emitters do strongly couple to the expected plasmonic modes (Fig. 1f, g). As both mode profiles cover the two QDs, the energy transfer rate between the QDs are expected to be enhanced. However, the enhanced energy transfer rate is still much smaller than the enhanced spontaneous emission rates and therefore the energy transfer efficiency is so low that no experimentally observable effect is expected (Supplementary Note 4).

## Discussion

In our study, AFM nanomanipulation enables the construction of the designed hybrid nanosystem. It has been widely used to produce plasmonic nanostructures[41,42] and couple individual emitting particles to plasmonic nanostructures[43–46]. It is ideal for proof-of-principle studies, since it allows for high-precision nanoassembly and importantly active tuning or optimization of individual structural parameters[41,43,47]. Photobleaching and chemical instability of QDs[48] are the biggest challenges to this

class of experiment and are mitigated with silica encapsulation in our experiment. If the QDs could be made very stable[48] or stable emitters of other types[49] are employed, the implementation of selective excitation and selective detection should be quite robust, although the optimal conditions (the excitation polarization for selective excitation and the polarizer angle for selective detection) and optimal selectivity may change a little with the fabricated structure parameters (Supplementary Note 1). Other fabrication methods could also be considered for practical fabrication of our plasmon–emitter hybrid nanosystem. For example, two-step electron beam lithography has been successfully applied to construct a variety of plasmon–emitter hybrid nanosystems[11,27,28,50]. It is a versatile method and can be utilized for repeated fabrication of our nanosystem. Self-assembly on DNA origami templates is also a potential fabrication approach for large-scale fabrication of our hybrid nanosystem[51,52].

In this experiment, the interaction between the emitters is incoherent. If the plasmon-mediated interaction between the

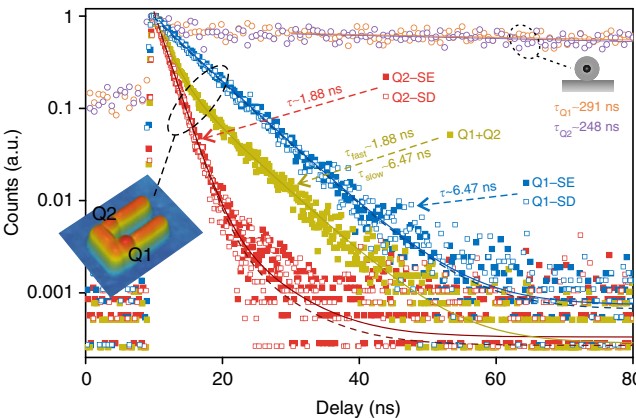

**Fig. 4** Lifetime measurements under different excitation and detection conditions. Q1-SE (blue solid data points and blue solid fitting curve): Q1 selectively excited, without selective detection; Q2-SE (red solid data points and red solid fitting curve): Q2 selectively excited, without selective detection; Q1+Q2 (yellow-green solid data points and yellow-green solid fitting curve): Q1 and Q2 equally excited, without selective detection; Q1-SD (blue hollowed data points and blue dashed fitting curve): Q1 and Q2 equally excited, Q1 selectively detected; Q2-SD (red hollowed data points and red dashed fitting curve): Q1 and Q2 equally excited, Q2 selectively detected. The lifetimes are remarkably reduced compared with that of Q1 (orange hollowed) and Q2 (purple hollowed) before they couple to the gold nanostructure. These lifetime measurements further confirm selective far-field excitation and detection, as well as indicate strong plasmon–emitter interactions

emitters is further enhanced by narrowing the gaps between GNRs (see the structure in Supplementary Fig. 15a, which is achievable in principal with AFM nanomanipulation), and if nearly identical quantum emitters with longer dephasing times (for instance, colour centres in nanodiamonds[35,49]) are employed, the plasmon-mediated interaction between the emitters can be made much faster than the dephasing processes between the emitters so that the emitters interact coherently with each other. Then plasmon-mediated entanglement[18–24] can take place in a U-shaped nanosystem and the demonstrated concept of plasmon-enabled far-field selective excitation and detection can be applied to make the entangled nanosystem writable (the original states of the emitters can be independently manipulated through selective excitation of the emitters) and readable (the quantum state of the system can be analysed through selective detection of the eigenstates), as theoretically discussed in the following (see Supplementary Note 8 for details). The quantum emitters are modelled as dipole emitters $\mu_1$ and $\mu_2$ without loss of generality. In such a coherent condition, the eigenstates of the singly excited system are the maximally entangled states $|\pm\rangle = (1/\sqrt{2})(|e_1,g_2\rangle \pm |g_1,e_2\rangle)$ (where $|g_i\rangle$ and $|e_i\rangle$ denote the ground and excited state of the emitter $\mu_i$). The entangled eigenstates $|\pm\rangle$ have distinct decay rates and distinct far-field radiation polarizations. The eigenstate $|+\rangle$ decays to plasmons with a fast decay rate $\gamma + \gamma_{12}$ (where $\gamma$ denotes decay rate induced by self emitter interaction while $\gamma_{12}$ denotes decay rate induced by mutual emitter interaction) and subsequently radiates to $x$-polarized photons, while the eigenstate $|-\rangle$ decays to plasmons with a slow decay rate $\gamma - \gamma_{12}$ and subsequently radiates to $y$-polarized photons. Starting with a singly excited unentangled initial state, for instance $|e_1,g_2\rangle$ (a superposition of eigenstates $|\pm\rangle$: $|e_1,g_2\rangle = (1/\sqrt{2})(|+\rangle + |-\rangle)$), which can be prepared by selective excitation of emitter $\mu_1$ (Supplementary Fig. 16), the large decay rate difference between the eigenstates $|\pm\rangle$ leads to spontaneous generation of entanglement by damping out the fast

decaying state $|+\rangle$ while leaving the slow decaying state $|-\rangle$. Since the eigenstates $|\pm\rangle$ have orthogonal radiation polarizations, they can be selectively detected through polarization analysis and the quantum state of the system can be analysed. For the coherently interacting emitters, the states of the emitters are entangled, and therefore the selective detection is to distinguish between the entangled states $|\pm\rangle$, rather than to distinguish between the emitters $\mu_{1,2}$.

Early reports have shown that user-specified nanoscopic optical field distributions in plasmonic nanostructures can be generated using femtosecond laser pulses with temporally shaped amplitude, phase and polarization states[53–55] or using spatially shaped polarized laser beams in the form of a coherent superposition of high-order Hermite–Gaussian beams[56], where temporal or spatial shaping parameters were obtained from evolutionary optimization algorithms[53,54], time-reversal algorithms[55] or optical inversion algorithms[56]. However, to the best of our knowledge, no experimental result with quantum emitters has been achieved yet with these coherent light control techniques, which are much more complicated than the present method. Our work represents an experimental proof-of-concept working in the relatively simple two-dimensional optical space of polarization. For selective excitations among more emitters, one could potentially generate more orthogonal optical states by introducing additional degrees of freedom to the excitation light. While for selective detection, one may use distinct radiation directions or distinct local field maps (probed, for instance, by fluorescent nanoparticles of distinct fluorescent wavelengths) to distinguish among multiple quantum emitters or quantum states. Combining such excitation and detection techniques, the platform we have developed is potentially scalable to more complicated quantum plasmonic nanosystems with multiple interacting quantum objects.

In conclusion, we have theoretically and experimentally shown that coupled quantum emitters in a plasmonic nanocavity can be selectively excited and detected from the far field, with their excitation cross-sections enlarged, radiation polarizations tailored and emission rates enhanced. In the deep subwavelength hybrid nanosystem, we exploit the capabilities of the plasmonic nanostructure as both optical nanocavities and optical receiving and transmitting nanoantennas throughout the process from excitation to decay and further to radiation. For excitation, the plasmonic nanostructure converts far-field excitation light into a desired local electric field distribution in a polarization-dependent manner, which allows far-field selective excitation of the emitters in close proximity. Further, the plasmonic nanostructure provides high LDOS to enhance the decay rates of the emitters (by up to ~132-fold) and to efficiently funnel the released energy from the excited emitters to their respective plasmonic modes in a location-dependent manner. Finally, the plasmonic nanostructure radiates the plasmonic modes to free space photons with distinct polarizations, which enables selective detection of the emitters from the far field. Our work represents a step forward in quantum nanophotonics by enabling selective addressing of coupled quantum emitters in deep subwavelength nanocavities, which may open up new degrees of freedom for light–matter interactions[57] and space–time-resolved pump–probe spectroscopies at the nanoscale[53] as well as promote the development of a broad class of plasmonic devices that will enable faster, more compact optics, communication and computation[31,32,58].

## Methods

**AFM nanomanipulation**. An Agilent 5500 SPM system in tapping mode is used for both imaging and pushing[59]. For imaging, the scanning parameters such as tapping amplitude, feedback gain and scan speed are optimized to produce high-quality images without moving the nanoparticles. For pushing, vibration amplitude

of the tapping tip is set smaller than half the height of the target nanoparticle and the feedback loop is turned off. Using tapping mode for pushing can not only reduce the possibility of noxious tip-particle adhesion but can also reduce wear of the AFM tip, which guarantees high-quality imaging and precise pushing during complicated nanoassembly without the need for time-consuming AFM probe replacement and concomitant sample relocation. To avoid long-time exposure of the QDs to the AFM laser during AFM imaging and manipulation, we choose AFM probes with an aluminum coating on the cantilever (Nanosensors PPP-NCHR) as well as move the AFM laser spot away from the front end of the cantilever where the tip resides.

**Sample fabrication.** Gold markers for co-localization of nano-objects (GNRs, QDs and hybrid nanosystems) by optical microscopy (darkfield light scattering microscopy and fluorescence microscopy) and AFM imaging are prepared on a silica glass substrate through photolithography and lift-off. Colloidal GNRs (Nanopartz Inc.) and silica-encapsulated[60] colloidal CdSeTe/ZnS core–shell QDs (Invitrogen, Qdot 800 ITK carboxyl) are then successively transferred to the substrate through spin-coating. The silica encapsulation facilitates AFM nanomanipulation (by increasing the physical size of QDs and protecting QDs from impact with the AFM tip and the substrate) and makes QDs on the substrate more stable (Supplementary Fig. 1). The constituent GNRs G1, G2 and G3 of the hybrid nanosystem are selected to have similar plasmonic responses with a resonance wavelength of ~715 nm according to their darkfield scattering spectra (Fig. 1c). The constituent QDs Q1 and Q2 are selected to have similar emission spectra with a central wavelength of ~808 nm and similar broadband excitation spectra (Fig. 1c). Through darkfield light scattering microscopy and fluorescence microscopy, the selected GNRs and QDs are optically localized with respect to the markers, which facilitates AFM localization. Finally, the selected GNRs and QDs are AFM localized and manipulated to assemble the hybrid nanosystem (Fig. 1b).

**Photoluminescence characterization.** We perform photoluminescence characterizations of the individual QDs and the hybrid nanosystem using a home-built microscopic single-molecule fluorescence detection system as shown in Supplementary Fig. 2, with an excitation module with automatic adaptive control over the wavelength, polarization and intensity of the excitation laser light.

The excitation laser light is generated and controlled as follows. A broadband pulse laser light is generated by pumping a nonlinear photonic crystal fiber (NKT Photonics, FemtoWHITE 800) with a 750 nm wavelength femtosecond pulse laser (Coherent, Mira 900). This broadband pulse laser light is subsequently filtered by a band-pass filter set (a long-pass edge filter and a short-pass edge filter) to generate a monochromatic pulsed laser light of a desired wavelength. The wavelength can be automatically selected or scanned by changing the filter set and tuning the incident angle of the filter. The polarization of the laser light is controlled with a linear polarizer, an achromatic half-waveplate (Thorlabs, SAHWP05M-1700) and an achromatic quarter-waveplate (Thorlabs, SAQWP05M-1700) to generate a purely polarized laser light of any elliptical polarization parameters $(\theta, \varphi)$. The intensity of the excitation laser light is controlled by a motorized variable neutral density filter. Finally, the excitation laser light is focussed to the sample by an achromatic focussing objective with low auto-fluorescence (Nikon, S Plan Fluor ELWD 40×, NA 0.6). The residual chromatic aberration is compensated by adjusting the height of the focussing objective for every wavelength point. The position of the focal spot is controlled by a pair of beam-steering mirrors. A beam displacement plate and the pair of beam-steering mirrors compensate for the beam displacement and beam angle change caused by changing or tuning of any optical elements in the beam path during automatic measurements.

The emitted photons from QDs are collected by an achromatic objective with low auto-fluorescence (Nikon, S Plan Fluor ELWD 60×, NA 0.7), passed through a long-pass filter to remove the excitation laser light, detected and analysed by the photon-counting module. In the photon-counting module, photons are detected by single photon detectors with an impulse response with a full-width at half-maximum of ~350 ps (Picoquant, tau-SPAD). The detector signals are counted by a photon counter (Stanford Research Systems, SR400) to get the emission intensity. A monochromator is inserted for emission spectra analysis. A linear polarizer is inserted for emission polarization analysis. A time-correlated single photon-counting module (Picoquant, Picoharp 300) is used for emission lifetime analysis.

The effective excitation/absorption cross-section of a QD is experimentally characterized as $\sigma = I_{em}/I_{ex}$, where the excitation intensity $I_{ex}$ is extracted from the power meter and the position of the variable neutral density filter, and the emission intensity $I_{em}$ is calculated based on the photon count and the photon collection efficiency (Supplementary Note 7). No absolute calibration is performed, therefore the measured excitation/absorption cross-sections are in arbitrary units. The excitation enhancement is defined as $\sigma_{QD+P}/\sigma_{QD}$, where $\sigma_{QD+P}$ and $\sigma_{QD}$ are the effective excitation/absorption cross-sections of the QD after and before coupling to the plasmonic nanostructure, respectively. Reference QDs are used to compensate any possible variations in the optical system once the sample has been re-loaded. To measure the excitation/absorption spectra (e.g. Fig. 1c) or excitation enhancement spectra (e.g. Fig. 3a,c), the excitation laser wavelength is scanned. During the scan, the emission intensity $I_{em}$ is set at a constant level through adaptive control of $I_{ex}$ for every excitation wavelength. The adaptive control of $I_{ex}$ is implemented through adjustment of the variable neutral density filter according

to the feedback from the photon counting. This adaptive excitation control guarantees both an adequate signal-to-noise ratio and a suitable excitation amplitude for the whole spectrum, which is especially important for QDs coupled to plasmonic nanostructures where the local field may have drastic spectral dependence, ranging from field enhancement to field suppression (e.g. Fig. 3a, c). Adaptive excitation control is also applied for scanning other excitation parameters, such as polarization parameters $\theta$ and $\varphi$ (e.g. Fig. 3d).

**Numerical simulation.** The numerical simulations are performed using a commercial finite-difference time-domain (FDTD) solver (FDTD solutions, Lumerical). To simulate elliptically polarized excitation with polarization parameters $\theta$ and $\varphi$, two separate simulations are performed with $x$-polarized ($\leftrightarrow$) and $y$-polarized ($\updownarrow$) excitation to obtain fields $\mathbf{E}^{\leftrightarrow}$ and $\mathbf{E}^{\updownarrow}$, which are then combined as $\cos\theta \cdot \mathbf{E}^{\leftrightarrow} + e^{i\varphi}\sin\theta \cdot \mathbf{E}^{\updownarrow}$. To simulate the far-field polarization, we simulate the near field distribution in a plane slightly below the plasmonic nanostructure and perform a far-field projection routine included in the FDTD software. See Supplementary Note 1 for the structural parameters used for the simulations. The optical constants of gold used for FDTD simulation are taken from ref. [61].

**Data availability.** The data that support the plots within this paper and other findings of this study are available from the corresponding author upon reasonable request.

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

## Acknowledgements

The authors would like to thank Yungui Ma (Zhejiang University) for his helpful discussion. This work was supported by the National Key Research and Development Program of China (2017YFA0205700), the Guangdong Innovative Research Team Program (201001D0104799318), the National Natural Science Foundation of China (11621101, 91233208), the Fundamental Research Funds for the Central Universities (2017FZA5001, 2018FZA5001), the China Postdoctoral Science Foundation (2017M621920, 2017M622722) and the Science and Technology Department of Zhejiang Province.

## Author contributions

J.T., J.X. and S.H. conceived the ideas for this research. J.T. and J.X. designed the structure and performed the numerical simulations. J.T., M.F., J.X. and F.B. fabricated the sample. J.T., J.X., F.B., M.F. and G.C. set up the optical system. J.T., M.F., J.X. and G.C. carried out the optical measurements. J.T. and J.S. did the theoretical study on entanglement. J.T., J.X., J.E. and S.H. wrote the manuscript. S.H. supervised the research. All authors discussed the manuscript and agreed on its final content.

## Additional information

**Competing interests:** The authors declare no competing interests.

