## [Peer Review File · Nature Communications]

REVIEWERS' COMMENTS:

Reviewer #2 (Remarks to the Author):

The article deals with the very important topic of addressing quantum emitters selectively using far field methods on a very much nanometric system. The system comprises 2 emitters and 3 metallic nanostructures. The nanostructures are in fact nanocavities that can be used to address each of the two emitters individually via their interaction with the plasmons from from the metallic nanostructures.

The article is sound and fairly clear, this is certainly not an easy task to assemble together these systems. What is even more remarkable is to have found the right conditions of excitation to be able to excite one emitter only and suppress the other one.

The paper is worth publishing although few points to answer and to work on:

1-the title does not seem the best one possible

2-the english could be revisited. The use of 'expedited' for instance does not seem appropriate

3-one can wonder how one gets to have conditions of almost perfect orthogonality in emission of the 2 emitters. It is not clear whether this was seeked for and it just happened od it has to happen for symmetry reasons and what not

4-at the end, the authors mention the conditions for creating entanglement and this is interesting but not directly relevant to the article. I think the authors should rephrase it differently by opening up to what one could do with different experimental conditions. Do the authors intend to work on this goal or is it too difficult experimentally?

5-the scheme to realise the addressing of 1 emitter only is rather complicated (see SI). The authors mention one could do more emitters. I would assume that this would require even more complicated interference conditions. Can the authors elaborate a bit more on this feasibility of such schemes?

If these points are addressed then the paper is most likely worth published in Nature Communications.

Manuscript No. NCOMMS-18-01426A

Original Title: Selectively addressing quantum emitters in a plasmonic nanocavity from the far field

Modified Title: Selective far-field addressing of coupled quantum dots in a plasmonic nanocavity

Responses to reviewer's comments

We appreciate very much the reviewer's positive evaluation and valuable suggestions on our work. In the following, we provide a point-by-point response to the comments and suggestions, together with the corresponding changes in the manuscript. As below, the reviewer's comments are written in **Green** and our responses in **blue**. The changes to the manuscript are given after the response in **red**.

Reviewer #2:

The article deals with the very important topic of addressing quantum emitters selectively using far field methods on a very much nanometric system. The system comprises 2 emitters and 3 metallic nanostructures. The nanostructures are in fact nanocavities that can be used to address each of the two emitters individually via their interaction with the plasmons from the metallic nanostructures.

The article is sound and fairly clear, this is certainly not an easy task to assemble together these systems. What is even more remarkable is to have found the right conditions of excitation to be able to excite one emitter only and suppress the other one.

The paper is worth publishing although few points to answer and to work on:

Response:

We thank the reviewer very much for the positive comments on our work.

1- The title does not seem the best one possible

Response:

We thank the reviewer for the advice and we have modified the title to "Selective far-field addressing of coupled quantum dots in a plasmonic nanocavity", which seems the best one possible.

Changes in the manuscript:

(1) We have changed the title of the manuscript from "Selectively addressing quantum emitters in a plasmonic nanocavity from the far field" to

"Selective far-field addressing of coupled quantum dots in a plasmonic nanocavity"

2- The English could be revisited. The use of 'expedited' for instance does not seem appropriate

Response:

As suggested, "expedited" has been changed to "enhanced" and other minor linguistic revisions have been made.

Changes in the manuscript:

(1) The word "expedited" has been changed to "enhanced" in the following sentences (with changes marked in red).

(Lines 21-24) “When we selectively excite or detect either emitter, we observe photon emission predominantly from the target emitter with up to 132-fold Purcell-enhanced emission rate, indicating individual addressability and strong plasmon-exciton interactions.”

(Lines 82-85) “Without coherence between the QDs in the current experimental system, the enhanced self interaction enhances spontaneous emission rate (known as the Purcell effect; Supplementary Note 3), while the enhanced mutual interaction enhances Förster energy transfer rate between the QDs (Supplementary Note 4).”

(Lines 257-259) “In conclusion, we have theoretically and experimentally shown that coupled quantum emitters in a plasmonic nanocavity can be selectively excited and detected from the far field, with their excitation cross sections enlarged, radiation polarizations tailored and emission rates enhanced.”

(2) The expression “accelerate the decay dynamics” has been changed to “enhance the decay rates” in the following sentence.

(Lines 264-267) “Further, the plasmonic nanostructure provides high LDOS to enhance the decay rates of the emitters (by up to ~132-fold) and to efficiently funnel the released energy from the excited emitters to their respective plasmonic modes in a location-dependent manner.”

(3) The word “owing” has been changed to “due” in the following sentences.

(Lines 32-33) “Due to these superior properties, plasmon-emitter hybrid nanosystems hold great promise as testbeds and building blocks for quantum optics and informatics.”

(Lines 64-65) “Due to the topology of the nanostructure, both x- and y-polarized illuminations predominantly generate y-oriented enhanced local fields at the QDs (Supplementary Fig. 5).”

(4) The words “to each other” or “with each other” following the word “orthogonal” have been deleted.

(Lines 140-142) “The measured polarizations of the emissions from the selectively excited QDs are nearly linear and roughly orthogonal (Fig. 3b), which facilitates far-field selective detection.”

(Lines 533-534) “The red pulses illustrate the photons emitted from each QD with plasmon-tailored polarizations (nearly linear and roughly orthogonal).”

3- One can wonder how one gets to have conditions of almost perfect orthogonality in emission of the 2 emitters. It is not clear whether this was sought for and it just happened or it has to happen for symmetry reasons and what not.

Response:

The rough orthogonality in the polarizations of the emissions from the 2 emitters is a result of the U-shaped design. The symmetry in the structure is one of the reasons but not sufficient for it to happen.

As we stated in the manuscript, “the three constituent GNRs G1, G2 and G3 behave like three linearly polarized electric dipoles with respective orientations, amplitudes and phases, which combine to form an elliptically polarized effective electric dipole”.

For symmetry reasons, the effective electric dipole for Q1 emission and that for Q2 emission are mirror-symmetric with respect to the y-axis. However, this is not enough for them to be roughly orthogonal.

For them to be roughly orthogonal, the effective electric dipole for Q1 emission must be oriented at an angle of around 45° and that for Q2 must be oriented at an angle of around 135° . For this to happen, the x- and y-components of the effective electric dipole must have similar amplitudes. This is guaranteed in our U-shaped design. Since the GNRs are similar and the emitter is in the middle of the gap between the two nearby orthogonally oriented GNRs, the amount of energy coupled from the emitter to the two nearby GNRs should be similar, and therefore the x- and y-components of the effective electric dipole should have similar amplitudes.

We have revised the manuscript as follows to add this explanation for the orthogonality in emission polarizations.

Changes in the manuscript:

(1) We have added the explanation for polarization orthogonality to the main text as follows:

(Lines 95-114) “The three constituent GNRs G1, G2 and G3 behave like three **linearly polarized** electric dipoles with respective orientations, amplitudes and phases (see the electric displacement vectors in Fig. 1f and g), which combine to form an elliptically polarized effective electric dipole, with its x-component contributed by G3 and its y-component contributed by G1 and G2. **For both modes in Fig. 1f and g, since the GNRs are similar and the emitter is in the middle of the gap between the two nearby orthogonally oriented GNRs, the amount of energy coupled from the emitter to the two nearby GNRs should be similar, and therefore the x- and y-components of the effective electric dipole should have similar amplitudes.** The key distinction between the two modes in Fig. 1f and g is the phase relation between the x- and y-components **of the effective electric dipole**. The x- and y-components **of the effective electric dipole** in Fig. 1f **(g)** are **roughly in-phase (anti-phase) and of similar amplitude, which should therefore combine to produce a nearly linearly polarized effective electric dipole oriented at an angle of around 45° (135°).** **These two roughly orthogonal effective electric dipoles should produce far-field radiations with roughly orthogonal polarizations.** Through far-field projections of the simulated near fields of the respective plasmonic modes, the polarization states **of the far-field radiations from Q1 and Q2** can be **precisely** obtained as shown in Fig. 1h and i. **The far-field radiation from Q1 (Q2) is linearly polarized with a degree of linear polarization (DOLP) of 0.98 (0.986) and a polarization angle of 41° (137°).** **Since the far-field radiations from Q1 and Q2 are nearly linearly polarized and their polarization angles are roughly orthogonal, we can selectively detect the emission from either QD with high transmittance by simply blocking the emission from the other QD using a linear polarizer.”**

4- At the end, the authors mention the conditions for creating entanglement and this is interesting but not directly relevant to the article. I think the authors should rephrase it differently by opening up to what one could do with different experimental conditions. Do the authors intend to work on this goal or is it too difficult experimentally?

Response:

We thank the reviewer for the very nice advice. We have rephrased it as suggested. In the rephrased discussion, we start with what one can do with certain experiment conditions: (Lines 212-223) “If the plasmon-mediated interaction between the emitters is further enhanced by narrowing the gaps between GNRs (see the structure in Supplementary Fig. 15a, which is achievable with AFM nanomanipulation), and if nearly identical quantum emitters with longer dephasing times (for instance, colour centers in diamond) are employed, the plasmon-mediated interaction between the emitters can be made much faster

than the dephasing processes between the emitters so that the emitters interact coherently with each other. Then plasmon-mediated entanglement can take place in our U-shaped nanosystem and the demonstrated concept of plasmon-enabled far-field selective excitation and detection can be applied to make the entangled nanosystem writable (the original states of the emitters can be independently manipulated through selective excitation of the emitters) and readable (the quantum state of the system can be analyzed through selective detection of the eigenstates), as theoretically discussed in the following.”

This is a challenging but promising direction and we indirectly indicate our intent to extend this work and hopefully achieve this goal with AFM nanomanipulation for fabrication and colour centers in nanodiamonds for nearly identical quantum emitters with longer dephasing times.

Changes in the manuscript:

(1) As suggested, we have rephrased the discussion on entanglement in the revised manuscript as follows:

(Lines 213-240) “In this experiment, the interaction between the emitters is incoherent. If the plasmon-mediated interaction between the emitters is further enhanced by narrowing the gaps between GNRs (see the structure in Supplementary Fig. 15a, which is achievable with AFM nanomanipulation), and if nearly identical quantum emitters with longer dephasing times (for instance, colour centers in nanodiamonds^{36,50}) are employed, the plasmon-mediated interaction between the emitters can be made much faster than the dephasing processes between the emitters so that the emitters interact coherently with each other. Then plasmon-mediated entanglement¹⁸⁻²⁴ can take place in a U-shaped nanosystem and the demonstrated concept of plasmon-enabled far-field selective excitation and detection can be applied to make the entangled nanosystem writable (the original states of the emitters can be independently manipulated through selective excitation of the emitters) and readable (the quantum state of the system can be analyzed through selective detection of the eigenstates), as theoretically discussed in the following (see Supplementary Note 8 for details). The quantum emitters are modelled as dipole emitters μ_1 and μ_2 without loss of generality. In such a coherent condition, the eigenstates of the singly excited system are the maximally entangled states $|\pm\rangle = (1/\sqrt{2})(|e_1, g_2\rangle \pm |g_1, e_2\rangle)$ (where $|g_i\rangle$ and $|e_i\rangle$ denote the ground and excited state of the emitter μ_i). The entangled eigenstates $|\pm\rangle$ have distinct decay rates and distinct far-field radiation polarizations. The eigenstate $|+\rangle$ decays to plasmons with a fast decay rate $\gamma + \gamma_{12}$ (where γ denotes decay rate induced by self emitter interaction while γ_{12} denotes decay rate induced by mutual emitter interaction) and subsequently radiates to x-polarized photons, while the eigenstate $|-\rangle$ decays to plasmons with a slow decay rate $\gamma - \gamma_{12}$ and subsequently radiates to y-polarized photons. Starting with a singly excited unentangled initial state, for instance $|e_1, g_2\rangle$ (a superposition of eigenstates $|\pm\rangle : |e_1, g_2\rangle = (1/\sqrt{2})(|+\rangle + |-\rangle)$) that can be prepared by selective excitation of emitter μ_1 (Supplementary Fig. 16), the large decay rate difference between the eigenstates $|\pm\rangle$ leads to spontaneous generation of entanglement by damping out the fast decaying state $|+\rangle$ while leaving the slow decaying state $|-\rangle$. Since the eigenstates $|\pm\rangle$ have orthogonal radiation polarizations, they can be selectively detected through polarization analysis and the quantum state of the system can be analyzed. For the coherently interacting emitters, the states of the emitters are entangled, and therefore the selective detection is to distinguish between the entangled states $|\pm\rangle$, rather than to distinguish between the emitters $\mu_{1,2}$.”

5- The scheme to realise the addressing of 1 emitter only is rather complicated (see SI). The authors mention one could do more emitters. I would assume that this would require even more complicated interference conditions. Can the authors elaborate a bit more on this feasibility of such schemes?

Response:

To elaborate a bit more on selective addressing among more emitters, we have expanded the relevant paragraph to the following one (with changes marked in red):

(Lines 241-256) “Early reports have shown that user-specified nanoscopic optical field distributions in plasmonic nanostructures can be generated using femtosecond laser pulses with temporally shaped amplitude, phase and polarization states⁵³⁻⁵⁵ or using spatially shaped polarized laser beams in the form of a coherent superposition of high-order Hermite-Gaussian beams⁵⁶, where temporal or spatial shaping parameters were obtained from evolutionary optimization algorithms^{53,54}, time-reversal algorithms⁵⁵ or optical inversion algorithms⁵⁶. However, to the best of our knowledge, no experimental result with quantum emitters has been achieved yet with these coherent light control techniques, which are much more complicated than the present method. Our work represents an experimental proof of concept working in the relatively simple two-dimensional optical space of polarization. For selective excitations among more emitters, one could potentially generate more orthogonal optical states by introducing additional degrees of freedom to the excitation light. While for selective detection, one may use distinct radiation directions or distinct local field maps (probed, for instance, by fluorescent nanoparticles of distinct fluorescent wavelengths) to distinguish among multiple quantum emitters or quantum states. Combining such excitation and detection techniques, the platform we have developed is potentially scalable to more complicated quantum plasmonic nanosystems with multiple interacting quantum objects.”

If these points are addressed then the paper is most likely worth published in Nature Communications.

Response:

We thank the reviewer again for the positive comments and valuable suggestions for improving our manuscript.